# Efficacy of Pilates on Pain, Functional Disorders and Quality of Life in Patients with Chronic Low Back Pain: A Systematic Review and Meta-Analysis

**DOI:** 10.3390/ijerph20042850

**Published:** 2023-02-06

**Authors:** Zhengze Yu, Yikun Yin, Jialin Wang, Xingxing Zhang, Hejia Cai, Fenglin Peng

**Affiliations:** 1College of Physical Education and Health, Guangxi Normal University, Guilin 541006, China; 2College of Physical Education and Health, Geely University of China, Chengdu 641432, China; 3Institute of Sports Medicine and Health, Chengdu Sport University, Chengdu 610041, China

**Keywords:** chronic low back pain, Pilates, pain relief, functional disorder, meta-analysis

## Abstract

Background: Chronic low back pain (CLBP) is a common health problem. Pilates is a unique exercise therapy. This meta-analysis aims to evaluate the efficacy of Pilates on pain, functional disorders, and quality of life in patients with chronic low back pain (CLBP). Methods: PubMed, Web of Science, CNKI, VIP, Wanfang Data, CBM, EBSCO, and Embase were searched. Randomized controlled trials of Pilates in the treatment of CLBP were collected based on the inclusion and exclusion criteria. The meta-analysis was performed using RevMan 5.4 and Stata 12.2. Results: 19 randomized controlled trials with a total of 1108 patients were included. Compared with the controls, the results showed the following values: Pain Scale [standard mean difference; SMD = −1.31, 95%CI (−1.80, −0.83), *p* < 0.00001], Oswestry Disability Index (ODI) [mean difference; MD = −4.35, 95%CI (−5.77, −2.94), *p* < 0.00001], Roland–Morris Disability Questionnaire (RMDQ) [MD = −2.26, 95%CI (-4.45, −0.08), *p* = 0.04], 36-item Short-Form (SF-36) (Physical Function (PF) [MD = 5.09, 95%CI (0.20, 9.99), *p* = 0.04], Role Physical (RP) [MD = 5.02, 95%CI (−1.03, 11.06), *p* = 0.10], Bodily Pain (BP) [MD = 8.79, 95%CI (−1.57, 19.16), *p* = 0.10], General Health (GH) [MD = 8.45, 95%CI (−5.61, 22.51), *p* = 0.24], Vitality (VT) [MD = 8.20, 95%CI(−2.30, 18.71), *p* = 0.13], Social Functioning (SF) [MD = −1.11, 95%CI (−7.70, 5.48), *p* = 0.74], Role Emotional (RE) [MD = 0.86, 95%CI (−5.53, 7.25), *p* = 0.79], Mental Health (MH) [MD = 11.04, 95%CI (−12.51, 34.59), *p* = 0.36]), Quebec Back in Disability Scale (QBPDS) [MD = −5.51, 95%CI (−23.84, 12.81), *p* = 0.56], and the sit-and-reach test [MD = 1.81, 95%CI (−0.25, 3.88), *p* = 0.09]. Conclusions: This meta-analysis reveals that Pilates may have positive efficacy for pain relief and the improvement of functional disorders in CLBP patients, but the improvement in quality of life seems to be less obvious. Registration: PROSPERO CRD42022348173.

## 1. Introduction

Low back pain (LBP) has been a prevalent health problem for adults. The incidence is as high as 84%. In modern society, with the accelerated pace of life and the increased pressure of work, the incidence of LBP is increasing year by year. From 2006 to 2016, the incidence of LBP increased by 18% [1]. Typically, LBP is clinically characterized by pain at the lower costal margin, lumbosacral region, and buttock region, with the potential for radiating pain to the lower extremities [2]. From 1990 to 2015, the years lived with disability caused by LBP increased by 54% [3]. The lifetime prevalence of LBP is approximately 70% to 80%. About 10% to 20% of LBP patients experienced pain lasting at least 3 months and progressed to chronic low back pain (CLBP). Clinically, CLBP is treated in a variety of ways. Major non-surgical treatment methods for CLBP include pharmacotherapy, physiotherapy, and exercise therapy [4,5]. However, pharmacotherapy may cause nausea, constipation, tiredness, and other side effects, and it is difficult for physiotherapy to relieve long-term pain [6,7,8]. In recent years, exercise therapy has been a preferred treatment method because of its characteristics of minimal harm, low cost, and ease of implementation [9].

Pilates, as an exercise therapy, is widely used in clinical rehabilitation. Pilates has been proven to have positive effects in relieving shoulder–neck discomfort and low back pain, enhancing joint mobility, improving physical balance ability, reducing the risk of falling in the elderly, and so on [10,11,12,13]. Pilates is a unique training system that was created in the early 20th century by a German named Joseph Pilates [14]. Pilates mainly trains deep core muscles, including transversus abdominis, diaphragm, abdominal oblique muscles, multifidus, and pelvic floor muscles, to enhance core muscles strength and endurance, maintain and improve somatic motor nerve control, increase spinal control, and improve somatic stability [15,16,17].

Some clinical studies have found that Pilates had positive effects on pain relief and improvement of functional disability in CLBP patients [18,19,20], while some other studies showed that it was not significantly different from routine rehabilitation training [21,22]. In addition, existing systematic reviews of RCTs have confirmed that Pilates provide better pain relief than minimal interventions in patients with chronic low back pain [23,24]. Another systematic review has confirmed that Pilates offers greater improvement in pain and functional ability compared to usual care and physical activity in the short term [25]. However, in these studies, only the minimal intervention, the usual care, and the routine physical activity were included in the integration process to the control groups of RCTs. This may have certain limitations for a complete demonstration of the benefits of Pilates in patients with chronic low back pain. Therefore, the objective of this systematic review and meta-analysis is to ascertain the efficacy of Pilates on pain, functional disorders, and quality of life in the treatment of patients with chronic low back pain. Additionally, it sought to ascertain whether Pilates can serve as a safe treatment method for patients with chronic low back pain.

## 2. Materials and Methods

### 2.1. Retrieval Strategy

This meta-analysis was planned and implemented according to the Preferred Reporting Items for Systematic Reviews and Meta-Analyses (PRISMA) guidelines [26]. The protocol was registered on the international prospective register of systematic reviews (http://www.crd.york.ac.uk/PROSPERO, accessed on 1 August 2022) with the registration number CRD42022348173.

In this study, PubMed, Web of Science, CNKI, VIP, Wanfang Data, CBM, EBSCO, and Embase were searched. The search time ranged from the date of database construction to November 2022. The last retrieval date is 20 November 2022. The literature search was conducted using a combination of subject terms and free terms. The search terms included “Pilates”, “Pilates training”, “low back pain”, “back pain”, “low back ache”, “nonspecific low back pain”, “chronic nonspecific low back pain”, “chronic nonspecific lumbago”, “chronic nonspecific lower back pain”, “chronic nonspecific lumbar pain”, and “non-specific lower back pain”. In order to obtain all the randomized control trials related to Pilates intervention in the treatment of CLBP patients, we also traced the references of the retrieved literature to supplement the relevant literature. The full search strategy for each database is presented in Appendix A.

### 2.2. Eligibility Criteria and Outcome Indicators

The eligibility criteria were as follows:Participants—CLBP patients (disease duration more than 3 months/12 weeks, aged 18–64 years), regardless of race and nationality, whose physical examination showed tenderness or pain in the lumbosacral region but no positive result in the straight leg raising test and the strengthening test, while excluding low back pain caused by other diseases, such as fracture, infection, and tumor, and ensuring no structural lesion in the lumbar spine in the imaging examination;Study design—randomized controlled trials (RCTs);Primary treatment methods—Pilates alone or in combination with other treatment methods;Treatment methods for the controls—any other treatment methods, including routine treatment, sham treatment, and no treatment;Literature data—the literature with complete data, which is able to effectively extract data and obtain original texts;Languages—the literature published in English or Chinese;Literature type—journal articles.The primary outcome indicators were as follows:Pain Scale, which was used to evaluate the pain intensity, including the Visual Analogue Scale (VAS) (ICC = 0.76–0.84) and Numerical Rating Scale (NRS) [27,28,29];Oswestry Disability Index (ODI). The ODI was used to evaluate lumbar vertebra function disorders in CLBP patients, consisting of 9 questions with 6 options per question, corresponding to 0 to 5 points, thus, giving a maximum score of 50, with a final score equal to actual score/45 × 100%. The higher the final score, the more severe the lumbar vertebra dysfunction (ICC = 0.99) [30,31];Roland–Morris Disability Questionnaire (RMDQ). The RMDQ was used to evaluate self-test dysfunction in CLBP patients. Scores ranged from 0 (no functional impairment) to 24 (severe functional impairment), with higher scores indicating more pronounced function disorders [32,33,34].The secondary outcome indicators were as follows:A 36-item Short-Form (SF-36). The SF-36 was used to evaluate the quality of life in CLBP patients, including eight dimensions of Physical Function (PF), Role Physical (RP), Bodily Pain (BP), General Health (GH), Vitality (VT), Social Functioning (SF), Role Emotion (RE), and Mental Health (MH). Scores ranged from 0 to 100, with higher scores indicating better quality of life (ICC > 0.85) [35];Quebec Back in Disability Scale (QBPDS). The QBPDS was used to assess fear of reinjury following a sports injury. The scale consists of 17 items with scores ranging from 17 (no fear) to 68 (highest fear), and the scale has good reliability and validity in CLBP patients [36,37,38,39];Sit-and-reach test. The test was used to evaluate hamstring tendon flexibility as well as lower back flexibility and lumbar extension (ICC = 0.94) [40,41].

### 2.3. Literature Screening and Data Extraction

Step 1—import the retrieved literature to the literature management software Endnotex9 (www.endnote.com). Step 2—exclude duplicate materials. Step 3—perform the first round of screening by reading titles and abstracts. Step 4—after downloading full texts, conduct the second round of screening to determine if the inclusion criteria were met.

Two independent reviewers, ZY and YY, conducted the literature screening and data extraction. Then, cross-checking was performed. When a possible disagreement occurred, we solved it through discussion or negotiation with a third independent reviewer, XZ. In the literature screening, we first read the title to exclude the irrelevant literature. Then, we further read the abstract and the full text to determine whether to include it. If necessary, we would contact the author of the original research by email or telephone to obtain the unconfirmed information.

The extracted data were as follows:General information of the included literature, namely the title, the first author, and the year of publication;General characteristics of the patients, namely the number of cases in each group, the age, and the duration of the disease;Treatment specifics and the follow-up time;Key elements of bias risk assessment;Focused outcome indicators.

### 2.4. Quality Assessment

Two independent reviewers used the Cochrane Collaboration tool to examine the risk of bias for the included studies [42,43], and cross-checking was conducted. A grading of the literature quality was performed according to the Jadad scale. A score of 1 to 3 was considered low quality, and a score of 4 to 7 was considered high quality. The grading was also conducted by two independent reviewers, with the opinions of a third independent reviewer being consulted in the event of any disagreement.

### 2.5. Statistical Analysis

The statistical analysis was based on RevMan5.4 (the Review Manager software 5.4, The Nordic Cochrane Center, The Cochrane Collaboration). If the results included in the literature were continuous variables and from the same assessment method, we used the mean difference (MD) and 95% confidence interval (CI) for statistics. If the results were not from the same assessment method, the standard mean difference (SMD) and 95% confidence interval (CI) were conducted. The *p*-value and the I^2^ index were used as indicators to assess the heterogeneity among studies. There was no heterogeneity between studies when *p* ≥ 0.10, while *p* < 0.10 indicates that there was heterogeneity between studies. The I^2^ index represented the degree of heterogeneity between studies. If I^2^ < 50%, it indicated that there was slight heterogeneity between the studies, and the fixed effect model was used for analysis. If I^2^ ≥ 50%, there was heterogeneity in the study, and the random effect model was used for analysis [44]. The α value was set at 0.05. Stata 12.0 software was used to conduct the publication bias analysis and sensitivity analysis of Begg’s test for the studies with more than five included outcome indicators. The threshold for statistical significance was set at *p* < 0.05. The safety analysis was conducted to confirm the safety of Pilates.

## 3. Results

The initial search resulted in a total of 537 studies, and 6 studies were selected in other ways. EndNote X9 was used to remove duplicate documents, and there were 416 studies left. After reading the titles and abstracts, 41 studies were selected. Then, after reading the full texts, 22 studies were discarded because they did not meet the inclusion and exclusion criteria, and 19 studies were finally included [12,18,19,20,21,22,23,36,45,46,47,48,49,50,51,52,53,54,55]. The process is shown in Figure 1.

### 3.1. Study Characteristics

A total of 1108 patients were included in the 19 RCTs. The average disease duration ranged from 86 days to 11.6 years. The sample size of each study ranged from 17 to 101 patients. The average intervention cycle is 6.8 weeks (ranging from 4 to 13 weeks), with training 3.1 times a week on average (ranging from 1 to 6 times per week). A total of 16 articles were published in the past 10 years (2013 to 2022), accounting for 84%. In 12 RCTs, the treatment method in trial groups was Pilates alone. In the remaining 7 RCTs, the treatment methods in trial groups were Pilates respectively combined with usual care, home exercise, physical therapy treatment, a standardized education program, infra-red radiation and back care, tendon puncture, the suspension training method, and non-steroidal anti-inflammatory drugs. For the controls, the treatment methods were conducted after the removal of Pilates in seven included RCTs, and there were six RCTs using the method of no treatment. Furthermore, usual care, home exercise, physical therapy treatment, as well as infra-red radiation and back care were, respectively, conducted in one, three, one, and one of the included RCTs. The details of the research characteristics are shown in Table 1 and Table 2.

Of the included 19 RCTs, 17 RCTs [12,18,19,20,21,22,23,36,45,48,49,50,51,52,53,54,55] conducted Pain Scale. Furthermore, ODI and RMDQ were used in nine [12,19,20,21,46,50,52,53,55] and eight [20,23,36,46,48,49,51,54] RCTs, respectively. In addition, SF-36, the sit-and-reach test, and QBPDS were separately performed in four [45,46,48,53], 3 [21,48,53] and two [47,53] RCTs.

The risk of bias assessment was performed using RevMan5.4 software, according to the *Cochrane Handbook for Systematic Reviews*. The results are shown in Figure 2 and Figure 3. The quality of the literature was graded according to the Jadad scale, with two studies judged to be of low quality and the remaining studies considered to be of high quality. The details are presented in Appendix A.

### 3.2. Results of Meta-Analysis

According to the results of the heterogeneity assessment, the random effects model meta-analysis was conducted to analyze the efficacy of Pilates in the context of the Pain Scale, ODI, and RMDQ of CLBP patients. The results showed that there were statistically significant differences between trial groups and control groups, which indicated that Pilates has a positive improvement on Pain Scale [SMD = −1.31, 95%CI (−1.80, −0.83), *p* < 0.00001], ODI [MD = −4.35, 95%CI(−5.77,−2.94), *p* < 0.00001], and RMDQ [MD = −2.26, 95%CI (−4.45, −0.08), *p* = 0.04] in CLBP patients. The results of the meta-analysis are presented in Figure 4, Figure 5 and Figure 6.

Four of the included RCTs [45,46,48,53] reported the effect of Pilates on four dimensions of SF-36, including Physical Function (PF), Role Physical (RP), Bodily Pain (BP), and General Health (GH), in the CLBP patients. The fixed effects model and random effects model were, respectively, used in the meta-analysis. The results showed the following: PF in trial groups was significantly different from control groups [MD = 5.09, 95%CI (0.20, 9.99), *p* = 0.04], and there was no statistically significant difference in RP [MD = 5.02, 95%CI (−1.03, 11.06), *p* = 0.10], BP [MD = 8.79, 95%CI (−1.57, 19.16), *p* = 0.10], and GH [MD = 8.45, 95%CI (−5.61, 22.51), *p* = 0.24] between trial groups and the controls. These results indicated that Pilates could positively improve SF-36-PF but not RP, BP, and GH in CLBP patients. The results of the meta-analysis are presented in Figure 7A–D. Three RCTs [45,46,48] involved the dimensions of Vitality (VT), Social Functioning (SF), Role Emotion, (RE) and Mental Health (MH). The meta-analysis showed that there was no statistically significant difference in VT [MD = 8.20, 95%CI (−2.30, 18.71), *p* = 0.13], SF [MD = −1.11, 95%CI (−7.70, 5.48), *p* = 0.74], RE [MD = 0.86, 95%CI (−5.53, 7.25), *p* = 0.79], and MH [MD = 11.04, 95%CI (−12.51, 34.59), *p* = 0.36] between trial groups and control groups, which suggested that Pilates had no positive effect on SF-36-VT, SF, RE, and MH in CLBP patients. The results of the meta-analysis are presented in Figure 7E–H.

Finally, meta-analyses for the efficacy of Pilates on QBPDS [MD = −5.51, 95%CI (−23.84, 12.81), *p* = 0.56] and the sit-and-reach test [MD = 1.81, 95%CI (−0.25, 3.88), *p* = 0.09] in CLBP patients were conducted. The results showed that there was a significant difference in the sit-and-reach test, but not QBPDS, between trial groups and control groups, which indicated that Pilates had a positive effect on the sit-and-reach test but not QBPDS in CLBP patients. The results of the meta-analysis are presented in Figure 8 and Figure 9.

### 3.3. Follow-Ups Analysis

To determine the long-term effects of Pilates on chronic low back pain, we performed a meta-analysis for Pain Scale, ODI, RMDQ, and SF-36 in follow-ups. A total of six RCTs followed the Pain Scale [12,23,45,48,50,54]; a random effects model meta-analysis showed that Pilates was superior to the controls in long-term pain relief in CLBP patients, the difference was statistically significant [MD = −0.70, 95%CI (−1.38, −0.03), *p* = 0.04]. In two RCTs [12,50], the follow-ups involved ODI. A random effects model meta-analysis showed that the decrease in ODI scores was more pronounced in CLBP patients after the treatment of Pilates compared with the controls, with a statistically significant difference [MD = −6.66, 95%CI (−13.12, −0.86), *p* = 0.03]. Furthermore, there were three RCTs [23,48,54] which reported RMDQ in follow-ups, and a fixed effects model meta-analysis showed that Pilates had a more positive effect on the decrease in RMDQ scores in CLBP patients in the comparison with the controls, with a statistically significant difference [MD = −1.97, 95%CI (−3.53, −0.40), *p* = 0.01]. For RCTs with follow-ups on SF-36, we included a total of two [45,48]. A fixed effects model meta-analysis showed that there was no statistically significant difference in all dimensions of SF-36 between the trial groups and control groups. The details are presented in Appendix A.

### 3.4. Publication Bias and Sensitivity Analysis

Begg’s test was conducted to analyze publication bias for the outcome indicators which were involved in five or more RCTs. The results showed that there was a publication bias risk for Pain Scale (t = −2.98, *p* = 0.008, *p* < 0.05), and there was no publication bias for ODI (t = −0.89, *p* = 0.405, *p* > 0.05) and RMDQ (t = −0.42, *p* = 0.686, *p* > 0.05). The details are presented in Appendix A.

We conducted a sensitivity analysis through one-by-one elimination for Pain Scale, ODI, and RMDQ in the included RCTs. After excluding any RCTs, there was no significant change in the pooled results and the results were stable. The details are presented in Appendix A.

### 3.5. Safety Analysis

Only one RCT [36] reported on safety and adverse events in patients after Pilates. In this study, the patient’s symptoms in the lumbar spine showed no aggravation, but we cannot determine the safety of Pilates.

## 4. Discussion

In this meta-analysis, we included 19 RCTs, with 1108 CLBP patients. The results showed that Pilates had a positive effect on Pain Scale, ODI, RMDQ and the sit-and-reach test, but that it had no obvious improvement on most dimensions of SF-36 and QBPDS. This suggested a beneficial effect of Pilates on the relief of pain and improvement of functional disability in patients with chronic low back pain, with little effect on the quality of life. Furthermore, the results of the follow-up analysis revealed that the effect of relieving pain and improving functional disability was still maintained in the future period after Pilates treatment. The results on the relief of pain in CLBP patients by Pilates were approximately consistent with those of three previous systematic reviews on Pilates in the treatment of CLBP mentioned in the text. However, one of the systematic reviews summarized that Pilates is no better than other types of exercise in reducing pain in the short term [23]. This negative result, which differs from this meta-analysis, may be caused by the difference in the included studies due to the difference in the inclusion and exclusion criteria. Moreover, this past systematic review may have included a smaller number of studies.

Usually, patients with acute low back pain would experience better improvement and relief in pain as well as dysfunction within 6 weeks [56]. However, acute low back pain would develop into chronic low back pain in approximately 40% of patients, and back pain, as well as functional disability, will persist for more than 12 weeks [6,57]. Furthermore, CLBP patients usually suffered from muscle atrophy in core stable muscle groups (erector spinae, quadratus lumborum, rectus abdominis, internal and external oblique, transversus abdominis and multifidus, etc.) and decreased muscle strength and muscle endurance. Over time, muscle group coordination to maintain strength and stability was dysregulated, with spinal motor flexibility decreased and postural control dysfunction appearing. In exercise, unnecessary compensatory phenomena occurred to maintain spinal stability, and the threshold of muscle fatigue was reduced, leading to rapid fatigue [58,59]. In addition, due to the neurological control disorder of CLBP patients, the local stabilizing muscles started relatively late during the body movement, and the muscle recruitment was delayed, which leads to a further decrease in spinal stability [60]. Long-term low back pain would also lead to core reflex inhibition and nervous system control disorders, resulting in lumbar stability decline [61,62,63]. In this way, patients with chronic low back pain fell into a vicious circle of “muscle atrophy-pain-activity restriction”.

Pilates is a unique training method in that it follows six important training principles, consisting of centering (i.e., activation of core muscle groups), concentration (i.e., cognitive attention when performing exercises), control (i.e., postural control management during exercises), precision (i.e., exercises with few repetitions but precise movements, emphasizing the quality of the exercise), breathing (i.e., the coordination of movements and breathings in exercises, promoting the activation of deep trunk muscles), and flow (i.e., smoothness during exercises and flowing transition between consecutive exercises) [14,15,64]. Pilates focuses on the activation of deep core muscles, spinal stabilization exercises and body postural controls [65], which in turn enhances the muscle strength and muscle endurance of deep muscle groups, increases the interaction between them and the whole muscle, reduces joint compression and changes pelvic inclination, improves postural control, and improves the stability of the spine, so that the spine is upright and neutral in the pelvis, and decreases the perceived force of the lumbar spine on the stimuli sent by nociceptors, thereby achieving pain relief [20,55]. It may be precisely, therefore, that Pilates may relieve the pain in patients with chronic low back pain. Furthermore, Pilates emphasizes both the local stability movement of the spine and the overall movement of the body, as well as strengthening the nerve control of the core muscle group and sending out the command of voluntary movement through the vertebral bundle to control the musculoskeletal system to correct body deviation in real time to maintain stability, gradually form the correct sensorimotor ability, and improve lumbar dysfunction [62,66,67]. Probably because of these, Pilates promotes the improvement of functional disorders in patients with chronic low back pain.

## 5. Conclusions and Suggestions

In view of the results of this meta-analysis, Pilates seems to have positive efficacy for pain relief and the improvement of functional disorders in patients with chronic low back pain, and the effects of treatment may be maintained for a period. However, there appears to be no positive effect on quality of life in CLBP patients. Furthermore, the safety of Pilates could not be determined.

It is suggested that a more uniform and standardized study design and treatment protocol should be established in future studies, while the doses of Pilates should also be intensively studied. Furthermore, measurements of objective instruments, such as electromyography and nuclear magnetic resonance, should also be added, in order to explore changes in neuromuscular regulation and imaging, and further verify the authenticity of the efficacy.

## 6. Limitations

In this meta-analysis, the study languages of the included RCTs were only Chinese and English, and the study sample sizes were relatively small, which may have biased the results.In the included RCTs, the exercise methods of Pilates were not all consistent, and the treatment methods of the controls were not all consistent.There was high heterogeneity among small parts of the literature, which may have caused some influences on the reliability of the meta-analysis.The bias risk in Pain Scale, according to the publication bias analysis, may be due to the difference in the pain assessment methods among studies, which may lead to an effect on the reliability of study results.

## Figures and Tables

**Figure 1 ijerph-20-02850-f001:**
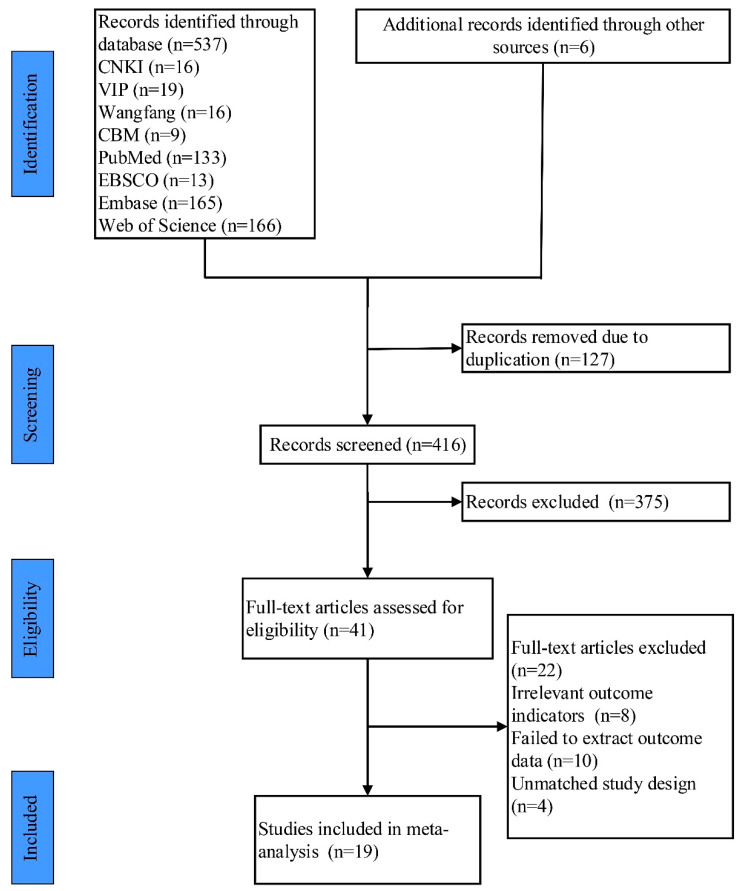
Study selection represented by a PRISMA flowchart.

**Figure 2 ijerph-20-02850-f002:**
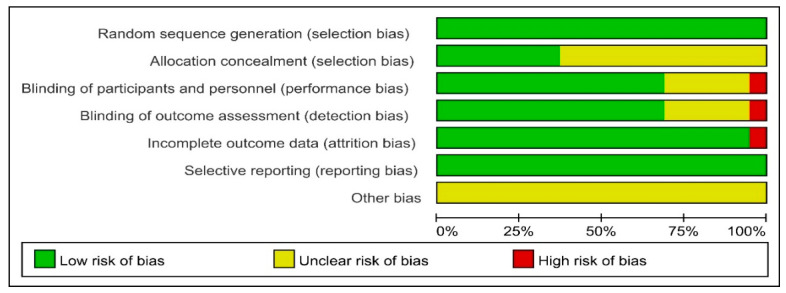
Résumé of results of risk of bias assessment.

**Figure 3 ijerph-20-02850-f003:**
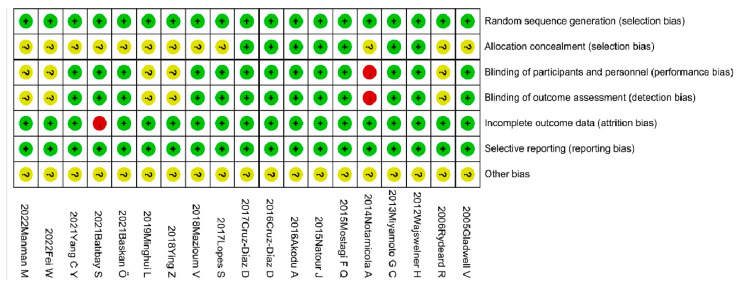
Punctuation of risk of bias tool [12,18,19,20,21,22,23,36,45,46,47,48,49,50,51,52,53,54,55].

**Figure 4 ijerph-20-02850-f004:**
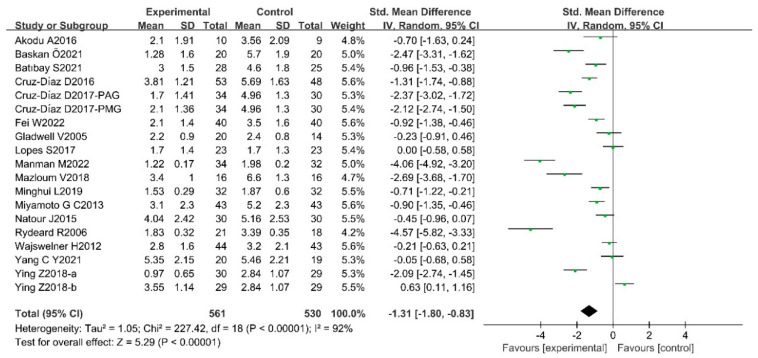
Meta-analysis of the effect of Pilates on Pain Scale in CLBP patients [12,18,19,20,21,22,23,36,43,45,48,49,50,51,52,54,55].

**Figure 5 ijerph-20-02850-f005:**
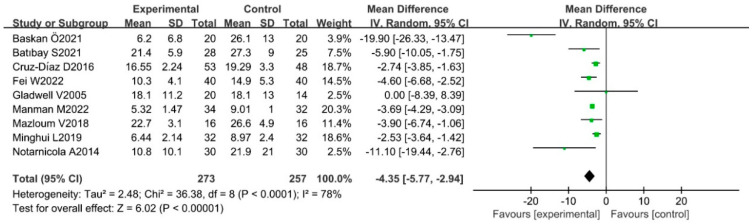
Meta-analysis of the effect of Pilates on ODI in CLBP patients [12,19,20,46,50,51,52,53,55].

**Figure 6 ijerph-20-02850-f006:**
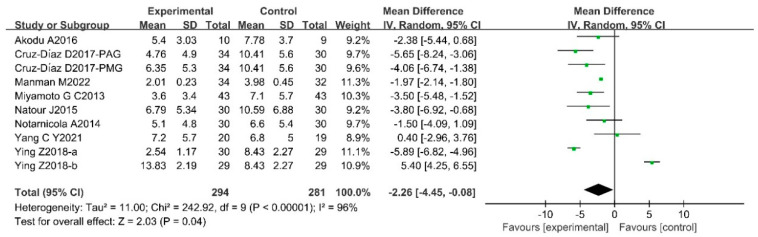
Meta-analysis of the effect of Pilates on RMDQ in CLBP patients [20,23,36,48,49,51,54].

**Figure 7 ijerph-20-02850-f007:**
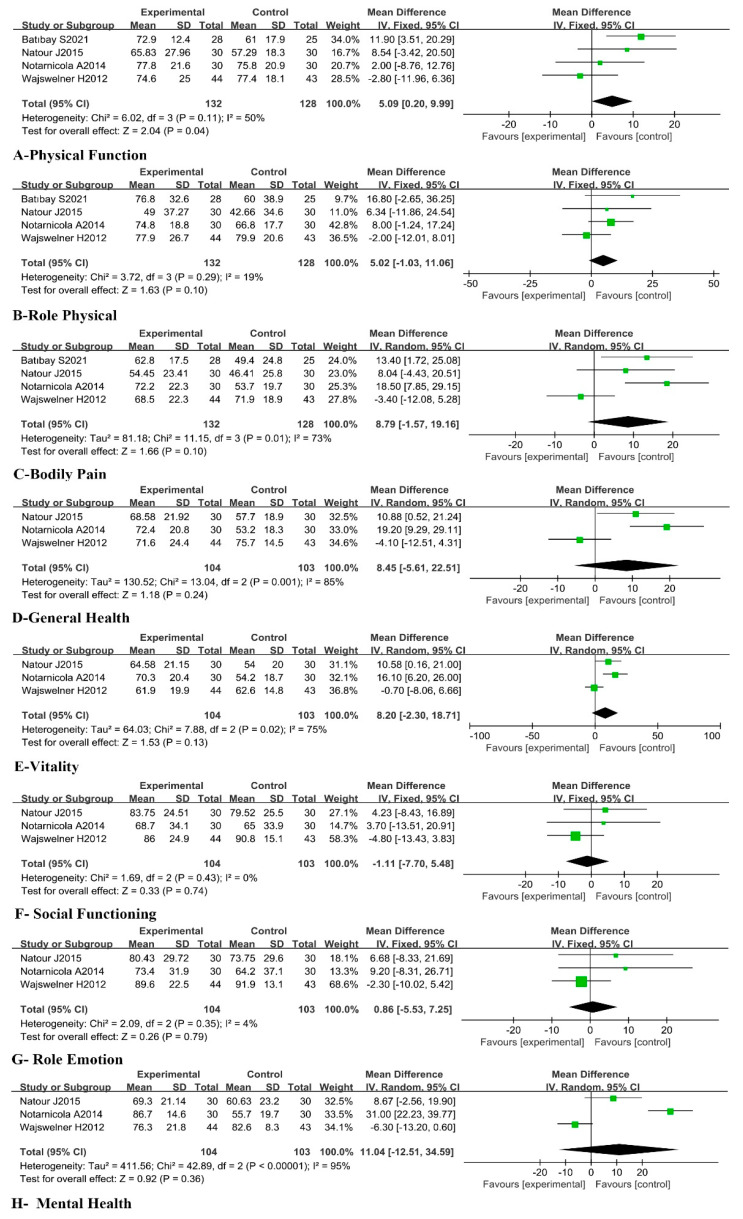
Meta-analysis of the effect of Pilates on eight dimensions of SF-36 in CLBP patients [45,46,48,53].

**Figure 8 ijerph-20-02850-f008:**
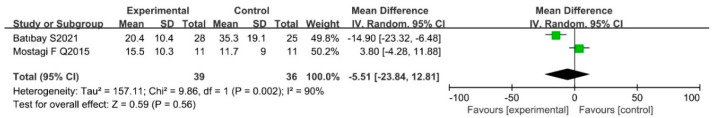
Meta-analysis of the effect of Pilates on QBPDS in CLBP patients [47,53].

**Figure 9 ijerph-20-02850-f009:**
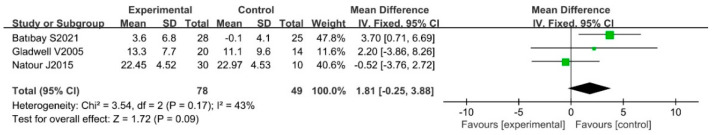
Meta-analysis of the effect of Pilates on the sit-and-reach test in CLBP patients [21,48,53].

**Table 1 ijerph-20-02850-t001:** The details of the research’s general characteristics.

Reference	Country	Sample Size (T/C)	Mean Age, Years (T/C)	Disease Duration
Gladwell, V. 2006 [21]	UK	20/14	36.9 ± 8.1/45.9 ± 8.0	9.6 ± 8.4 y/11.6 ± 12.3 y
Rydeard, R. 2006 [18]	Canada	18/21	37 ± 9/34 ± 8	5.5 y/9 y
Wajswelner, H. 2012 [45]	Australia	44/43	49.3 ± 14.1/48.9 ± 16.4	13.6 ± 14.2 y/14.2 ± 12.7 y
Miyamoto, G.C. 2013 [23]	Brazil	41/43	40.7 ± 11.8/38.3 ± 11.4	73.3 ± 79.6 m/56.7 ± 53.5 m
Notarnicola, A. 2014 [46]	Italy	30/30	46.9 ± 10.3/55.5 ± 7.1	96 ± 86.1 d/86 ± 89.6 d
Mostagi, F.Q. 2015 [47]	Brazil	10/7	36.1 ± 9/34.7 ± 8.1	-
Natour, J. 2015 [48]	Brazil	30/30	48.08 ± 12.98/47.79 ± 11.47	-
Akodu, A. 2016 [49]	Nigeria	10/10	45.30 ± 11.31/40.33 ± 14.5	-
Cruz-Díaz, D. 2016 [50]	Spain	53/48	69.57 ± 2.18/72.69 ± 3.53	-
Cruz-Díaz, D. 2017 [36]	Spain	PMG:34; PAG:34/30	PMG:36.94 ± 12.46; PAG:35.5 (11.98)/36.32 (10.67)	-
Lopes, S. 2017 [22]	Portugal	23/23	21.8 ± 3.2/22.8 ± 3.6	27.1 ± 16.6 m/31.0 ± 25.8 m
Mazloum, V. 2018 [12]	Iran	16/16	37.1 ± 9.5/39.3 ± 9.8	32.3 ± 18.3 m/32.4 ± 16.4 m
Ying, Z. 2018 [51]	China	a:30; b:29/29	a:36.29 ± 4.61; b:36.95 ± 4.40/36.25 ± 5.30	a:15.94 ± 5.08 m;b:14.98 ± 5.17 m/15.68 ± 5.23 m
Minghui, L. 2019 [52]	China	32/32	43.24 ± 11.54/45.16 ± 10.37	13.36 ± 3.44 m/12.12 ± 3.37 m
Baskan, Ö. 2021 [19]	Turkey	20/20	41.55 ± 3.39/38.95 ± 3.96	-
Batıbay, S. 2021 [53]	Turkey	28/25	49.3 ± 10.4/48.4 ± 9.3	5.8 ± 4.1 y/6.3 ± 3.5 y
Yang, C. 2021 [54]	China	20/19	50.5 ± 11.8/47.9 ± 15.9	-
Fei, W. 2022 [55]	China	40/40	37.4 ± 6.5/36.1 ± 7.7	3.4 ± 1.8 m/3.4 ± 1.8 m
Manman M.2022 [20]	China	34/32	44.21 ± 10.97 /44.39 ± 10.03	33.24 ± 11.01 m/32.35 ± 10.42 m

Abbreviations are as follows: t = time, d = day, w = week, m = month, y = year. Here, “-” indicates not mentioned. Furthermore, T = trial group, C = control group; a, tendon puncture combined with Pilates; b, Pilates alone; PAG, Pilates with apparatus group; PMG, Pilates mat group.

**Table 2 ijerph-20-02850-t002:** The details of research intervention and outcome indicators.

Reference	Treatment Methods		Dosage	Outcome	Follow-Up
	T	C			
Gladwell, V. 2006 [21]	Pilates	Ⅹ	60 min/t, 1 t/w, 6 w	①②③⑤	-
Rydeard, R. 2006 [18]	Pilates	Ⅰ	75 min/t, 3 t/w, 4 w	①	3 m, 6 m, 12 m
Wajswelner, H. 2012 [45]	Pilates	Ⅱ	60 min/t, 2 t/w, 6 w	①④	12 w, 24 w
Miyamoto, G.C. 2013 [23]	Pilates	Ⅹ	2 t/w, 6 w	①③	6 m
Notarnicola, A. 2014 [46]	Pilates	Ⅹ	60 min/t, 5 t/w, 6 m	②③④	-
Mostagi, F.Q. 2015 [47]	Pilates	Ⅲ	2 t/w, 8 w	⑥	3 m
Natour, J. 2015 [48]	Pilates and Ⅳ	Ⅳ	50 min/t, 2 t/w, 90 d	①③④⑤	180 d
Akodu, A. 2016 [49]	Pilates	Ⅴ	2 t/w, 4 w	①③	-
Cruz-Díaz, D. 2016 [50]	Pilates and Ⅲ	Ⅲ	2 t/w, 6 w	①②	1 y
Cruz-Díaz, D. 2017 [36]	PMG/PAG	Ⅹ	50 min/t, 2/w, 12 w	①③	-
Lopes, S. 2017 [22]	Pilates	Ⅹ	20 min/t	①	-
Mazloum, V. 2018 [12]	Pilates	Ⅹ	3 t/w, 6 w	①②	10 w
Ying, Z. 2018 [51]	a:Pilates and Ⅵ b:Pilates	Ⅵ	5 t/w, 8 w	①③	-
Minghui, L. 2019 [52]	Pilates and Ⅶ	Ⅶ	30 min/t, 5 t/w, 4 w	①②	-
Baskan, Ö. 2021 [19]	Pilates	Ⅱ	45 min/t, 3 times/w, 8 w	①②	-
Batıbay, S. 2021 [53]	Pilates	Ⅱ	60 min/t, 3 t/w, 8 w	①②④⑤⑥	-
Yang, C. 2021 [54]	Pilates and Ⅷ	Ⅷ	60 min/t, 2 t/w, 8 w	①③	26 w
Fei, W. 2022 [55]	Pilates and Ⅸ	Ⅸ	30 min/t, 6 t/w, 4 w	①②	-
Manman, M. 2022 [20]	Pilates and Ⅷ	Ⅷ	30 min/t, 5 t/w, 4 w	①②	-

Abbreviations are as follows: t = time, d = day, w = week, m = month, y = year; “-” indicates not mentioned; T = trial group, C = control group; a, tendon puncture combined with Pilates; b, Pilates alone; PAG, Pilates with apparatus group; PMG, Pilates mat group;Ⅰ, usual care; Ⅱ, home exercise; Ⅲ, physical therapy treatment; Ⅳ, standardized education program; Ⅴ, infra-red radiation and back care; Ⅵ, tendon puncture; Ⅶ, suspension training method, Ⅷ: non-steroidal anti-inflammatory drug; Ⅸ, massage; Ⅹ, no treatment. ① Pain Scale (VAS/NRS); ② Oswestry disability index, ODI; ③ Roland–Morris Disability Questionnaire, RMDQ; ④ 36-item Short-Form, SF-36; ⑤ sit-and-reach test; ⑥ Quebec Back in Disability Scale, QBPDS.

## Data Availability

The data are not publicly available for privacy reasons.

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
