# Peer review of "Efficacy of Pilates on Pain, Functional Disorders and Quality of Life in Patients with Chronic Low Back Pain: A Systematic Review and Meta-Analysis"

_ijerph, 2023, doi:10.3390/ijerph20042850_

Round 1

Reviewer 1 Report

First of all, congratulations for the research work done, then I will mention only one recommendation in order to get clearer and more precise information on your results. As far as methodology is concerned, I must congratulate you on your work.

- In the introduction section, it would be interesting to include these reviews as precedents and discuss the differences in the discussion section.

Miyamoto, G. C., Costa, L. O., & Cabral, C. M. (2013). Efficacy of the Pilates method for pain and disability in patients with chronic nonspecific low back pain: a systematic review with meta-analysis. Brazilian journal of physical therapy, 17(6), 517-532. https://doi.org/10.1590/S1413-35552012005000127

Lim, E. C., Poh, R. L., Low, A. Y., & Wong, W. P. (2011). Effects of Pilates-based exercises on pain and disability in individuals with persistent nonspecific low back pain: a systematic review with meta-analysis. The Journal of orthopaedic and sports physical therapy, 41(2), 70-80. https://doi.org/10.2519/jospt.2011.3393

Wells, C., Kolt, G. S., Marshall, P., Hill, B., & Bialocerkowski, A. (2014). The effectiveness of Pilates exercise in people with chronic low back pain: a systematic review. PloS one, 9(7), e100402. https://doi.org/10.1371/journal.pone.0100402

I hope that these bibliographic references may be useful and help you to further strengthen this manuscript.

Author Response

Dear reviewer,

We sincerely thank you for your important and valuable suggestion on our manuscript. We have revised our manuscript according to your suggestion. Thank you again for your hard work. We hope you can be satisfied with our response.

- In the introduction section, it would be interesting to include these reviews as precedents and discuss the differences in the discussion section.

Thank you for your suggestion. We have added the citation and revised the introduction and discussion in the revision of the manuscript.

Some clinical studies have found that Pilates had positive effects on pain relief and improvement of functional disability in CLBP patients [18-20] , while some other studies showed that it was not significantly different from routine rehabilitation training [21,22]. In addition, existing systematic reviews of RCTs have confirmed that Pilates provide better pain relief than minimal interventions in patients with chronic low back pain [23,24]. And another systematic review has confirmed that Pilates offers greater improvement in pain and functional ability compared to usual care and physical activity in the short term [25]. However, in these studies, only the minimal intervention, the usual care, and the routine physical activity were included in the integration process to the control groups of RCTs. This may have certain limitations for a complete demonstration of the benefits of Pilates in patients with chronic low back pain. Therefore, the objective of this systematic review and meta-analysis is to ascertain the efficacy of Pilates on pain, functional disorders and quality of life in the treatment of patients with chronic low back pain. Additionally, it sought to ascertain whether Pilates can serve as a safe treatment method for patients with chronic low back pain.

In this meta-analysis, we included 19 RCTs, with 1108 CLBP patients. The results showed that Pilates had a positive effect on Pain Scale, ODI, RMDQ and Sit-and-reach Test, but had no obvious improvement on most dimensions of SF-36 and QBPDS. This suggested a beneficial effect of Pilates on the relief of pain and improvement of functional disability in patients with chronic low back pain, with little effect on the quality of life. And the results of the follow-up analysis revealed that the effect of relieving pain and improving functional disability was still maintained in the future period after Pilates treatment. The results on the relief of pain in CLBP patients by Pilates were approximately consistent with those of three previous systematic reviews on Pilates in the treatment of CLBP mentioned in the text. However, one of the systematic reviews summarized that Pilates is no better than other types of exercise in reducing pain in the short term [23]. This negative result, which differs from this meta-analysis, may be caused by the difference in the included studies due to the difference in the inclusion and exclusion criteria. Moreover, this past systematic review may have included a smaller number of studies.

Reviewer 2 Report

Dear Authors: I have reviewed the manuscript entitled “Efficacy of Pilates on Pain, Functional Disorders and Quality of Life in Patients with Chronic Low Back Pain: A Meta-Analysis and Systematic Review” for consideration to be published in International Journal of Environmental Research and Public Health. It is a very interesting review that probably will encourage another authors to perform further studies. Congratulations to the authors”

Page 5, line 189 (table 1). The authors should reconsider making changes to table 1, either by separating the information into two tables, or by removing a column, since too much concentrated information generates some confusion when reading it.

Author Response

Dear reviewer,

We sincerely thank you for your important and valuable suggestion on our manuscript. We have revised our manuscript according to your suggestion. Thank you again for your hard work. We hope you can be satisfied with our response.

Page 5, line 189 (table 1). The authors should reconsider making changes to table 1, either by separating the information into two tables, or by removing a column, since too much concentrated information generates some confusion when reading it.

Thank you for your suggestion. We have divided the contents of Table 1 into two tables. The details have been presented in the revision of the manuscript.

Table 1. The details of research general characteristics.

Reference

Country

Sample size(T/C)

Mean age, years(T/C)

Disease Duration

Gladwell V.2005

UK

20/14

36.9±8.1/45.9±8.0

9.6±8.4y/11.6±12.3y

Rydeard R.2006

Canada

18/21

37±9/34±8

5.5y/9y

Wajswelner H.2012

Australia

44/43

49.3±14.1/48.9±16.4

13.6±14.2y/14.2±12.7y

Miyamoto G. C.2013

Brazil

41/43

40.7±11.8/38.3±11.4

73.3±79.6m/ 56.7±53.5m

Notarnicola A.2014

Italy

30/30

46.9±10.3/55.5±7.1

96±86.1d/86±89.6d

Mostagi F Q.2015

Brazil

10/7

36.1±9/34.7±8.1

-

Natour J.2015

Brazil

30/30

48.08±12.98/47.79±11.47

-

Akodu A.2016

Nigeria

10/10

45.30±11.31/40.33±14.5

-

Cruz-Díaz D.2016

Spain

53/48

69.57±2.18/72.69±3.53

-

Cruz-Díaz D.2017

Spain

PMG:34; PAG:34/30

PMG:36.94±12,46; PAG:35.5 (11.98)/36.32(10.67)

-

Lopes S.2017

Portugal

23/23

21.8±3.2/22.8±3.6

27.1±16.6m/31.0±25.8m

Mazloum V.2018

Iran

16/16

37.1±9.5/39.3±9.8

32.3±18.3m/32.4±16.4m

Ying Z.2019

China

a:30; b:29/29

a:36.29±4.61;

b: 36.95±4.40/36.25±5.30

a:15.94±5.08m;b:14.98±5.17m/ 15.68±5.23m

Minghui L.2019

China

32/32

43.24±11.54/45.16±10.37

13.36±3.44m/12.12±3.37m

Baskan Ö.2021

Turkey

20/20

41.55±3.39/38.95±3.96

-

Batıbay S.2021

Turkey

28/25

49.3±10.4/48.4±9.3

5.8±4.1y/6.3±3.5y

Yang C.2021

China

20/19

50.5±11.8/47.9±15.9

-

Fei W.2022

China

40/40

37.4±6.5/36.1±7.7

3.4±1.8m/3.4±1.8m

Manman M.2022

China

34/32

44.21±10.97

/44.39±10.03

33.24±11.01m/32.35±10.42m

Notes: t=time, d=day, w=week, m=month, y=year. "-": Not mentioned. T=trial group, C=control group. a: Tendon puncture combined with Pilates; b: Pilates alone. PAG: Pilates with apparatus group; PMG: Pilates Mat group.

Table 2. The details of research intervention and outcome indicators.

Reference

Treatment methods

Dosage

Outcome

Follow-up

T

C

Gladwell V.2005

Pilates

60min/t,1t/ w,6w

①②③⑤

-

Rydeard R.2006

Pilates

75min/t, 3t/ w,4w

3m,6m,12m

Wajswelner H.2012

Pilates

60min/t, 2t/w,6w

①④

12w,24w

Miyamoto G. C.2013

Pilates

2t/w,6w

①③

6m

Notarnicola A.2014

Pilates

60min/t, 5t/ w, 6m

②③④

-

Mostagi F Q.2015

Pilates

2t/w, 8w

3m

Natour J.2015

Pilates&Ⅳ

50min/t, 2t/w, 90d

①③④⑤

180d

Akodu A.2016

Pilates

2t/w, 4w

①③

-

Cruz-Díaz D.2016

Pilates&Ⅲ

2t/w,6w

①②

1y

Cruz-Díaz D.2017

PMG/PAG

50min/t, 2/w,12w

①③

-

Lopes S.2017

Pilates

20min/t

-

Mazloum V.2018

Pilates

3t/w,6w

①②

10w

Ying Z.2019

a:Pilates&Ⅵ

b:Pilates

5t/w,8w

①③

-

Minghui L.2019

Pilates&Ⅶ

30min/t, 5t/ w,4w

①②

-

Baskan Ö.2021

Pilates

45min/t, 3times/w,8w

①②

-

Batıbay S.2021

Pilates

60min/t, 3t/w,8w

①②④⑤⑥

-

Yang C.2021

Pilates&Ⅷ

60min/t, 2t/w,8w

①③

26w

Fei W.2022

Pilates&Ⅸ

30min/t, 6t/ w,4w

①②

-

Manman M.2022

Pilates&Ⅷ

30min/t, 5t/ w,4w

①②

-

Notes: t=time, d=day, w=week, m=month, y=year. "-" Not mentioned. T=trial group, C=control group. a: Tendon puncture combined with Pilates; b: Pilates alone. PAG: Pilates with apparatus group; PMG: Pilates Mat group. Ⅰ: Usual care; Ⅱ: Home exercise; Ⅲ: Physical therapy treatment; Ⅳ: Standardized education program; Ⅴ: Infra-red radiation and back care; Ⅵ: Tendon puncture; Ⅶ: Suspension training method; Ⅷ: non-steroidal anti-inflammatory drug; Ⅸ: Massage; Ⅹ: No treatment. ①Pain Scale (VAS/ NRS); ②Oswestry disability index, ODI; ③Roland-Morris Disability Questionnaire, RMDQ; ④36-item Short-Form, SF-36;⑤Sit-and-reach Test; ⑥Quebec Back in Disability Scale, QBPDS.

Reviewer 3 Report

Manuscript ID: ijerph-2129937

Title: Efficacy of Pilates on Pain, Functional Disorders and Quality of Life in Patients with Chronic Low Back Pain: A Meta-Analysis and Systematic Review

Authors: Zhengze Yu, Yikun Yin, Jialin Wang, Xingxing Zhang, Hejia Cai,
Fenglin Peng

Journal: International Journal of Environmental Research and Public Health

Dear authors,

I enjoyed reading your paper. Low back pain is defined as a public health issue. The study concerns an important problem of treatment of low back pain among adults. Exercise therapy is a preferred treatment method because of its characteristics of minimal harm, low cost, and ease of implementation. The objective of this meta-analysis is to ascertain the efficacy of Pilates in the treatment of patients with chronic low back pain. I have only some minor recommendations, which I will provide here.

 1. I propose a change in the title: Efficacy of Pilates on Pain, Functional Disorders and Quality of Life in Patients with Chronic Low Back Pain: A Systematic Review and Meta-Analysis

2. line 81: “in Supplemental Materials 1” should be changed to “in Supplementary Table 3”

Line 206: “in Supplemental Materials 2” should be changed to “in Supplementary Table 2”

3. Line 94: „Literature data” move to a new line.

4. Figure 1: insert spaces, for example: VIP(n=19) and it should be VIP (n=19).

5. Table 1: there should be a reference number instead of the year of publication, for example Gladwell et al. [21]. It will be easier to find the publication in the reference list.

6. Figure 3: the order of included studies should be the same as in Table 1.

7. Line 265, 335: there is Pilate, there should be Pilates.

8. Line 281: there is Supplementary Table S2, there should be Supplementary Table 1.

Supplementary Materials:

1. Table names should be unified with figure names, form example:

Supplementary Figure S1. And Supplementary Table S1.

2. I suggest changing the table order in Supplemental Materials according to the order in which they appear in the article

Supplementary Table S1. Search strategy for each database.

Supplementary Table S2. Jada scale score of included RTCs.

Supplementary Table S3. The pooled results of sensitivity analyses [MD(95% CI)].

3. Supplementary Table 2: explain below the table what 1, 2, 3 means.

4. Supplementary Table 3: The results presented in Table 3 are not clear. I suggest adding an additional column “Search”.

Search engine

Search

Search query

Date of search

Pubmed

#1

Pilates [Title/Abstract]

20-November-22

#3

#1 OR #2

5. Citation of publication number 66 is missing.

Despite the limitations, I congratulate the authors for their excellent work. I think the study should be accepted with minor changes and published by the International Journal of Environmental Research and Public Health.

Author Response

Dear reviewer,

We sincerely thank you for your important and valuable suggestions on our manuscript. We have revised our manuscript according to your suggestion. Thank you again for your hard work. The following are our point-by-point responses. We hope you can be satisfied with these.

  1. I propose a change in the title: Efficacy of Pilates on Pain, Functional Disorders and Quality of Life in Patients with Chronic Low Back Pain: A Systematic Review and Meta-Analysis

Thank you for your suggestion.

We have revised the title in the revision of the manuscript.

  1. line 81: “in Supplemental Materials 1” should be changed to “in Supplementary Table 3”

Line 206: “in Supplemental Materials 2” should be changed to “in Supplementary Table 2”

Thank you for your suggestion. Combined with the suggestions you mentioned later, we have revised the mistake in the revision of the manuscript.

The full search strategy for each database is presented in Supplemental Table S1.

The details are presented in Supplemental Table S2.

  1. Line 94: “Literature data” move to a new line.

Thank you for your suggestion. We have revised the mistake in the revision of the manuscript.

The eligibility criteria:

Participants: CLBP patients (disease duration more than 3 months/12 weeks, aged 18-64 years), regardless of race and nationality, whose physical examination showed tenderness pain in the lumbosacral region but no positive result in straight leg raising test and the strengthening test, with excluding from low back pain caused by other diseases such as fracture, infection and tumor, and no structural lesion in the lumbar spine in the imaging examination;

Study design: randomized controlled trials (RCTs);

Primary treatment methods: Pilates alone or in combination with other treatment methods;

Treatment methods for the controls: any other treatment methods, including routine treatment, sham treatment, and no treatment;

Literature data: literature with complete data, which is able to effectively extract data and obtain original texts;

Languages: literature published in English or Chinese;

Literature type: journal articles.

  1. Figure 1: insert spaces, for example: VIP(n=19) and it should be VIP (n=19).

Thank you for your suggestion. We have revised the mistake in the revision of the manuscript.

  1. Table 1: there should be a reference number instead of the year of publication, for example Gladwell et al. [21]. It will be easier to find the publication in the reference list.

Thank you for your suggestion. After considering the suggestions of you and another reviewer, we finally decided to divide the contents of Table 1 into two tables. The details have been presented in the revision of the manuscript.

  1. Figure 3: the order of included studies should be the same as in Table 1.

Thank you for your suggestion. We have changed the order in Figure 3 in the revision of the manuscript.

  1. Line 265, 335: there is Pilate, there should be Pilates.

Thank you for your suggestion. We have revised the mistake in the revision of the manuscript.

And there were 3 RCTs [43,46,52] reported RMDQ in follow-ups, a fixed effects model meta-analysis showed that Pilates had a more positive effect on the decrease of RMDQ scores in CLBP patients in the comparison with the controls, with a statistically significant difference [MD=-1.97,95%CI(-3.53,-0.40),P=0.01]. For RCTs with follow-ups on SF-36, we included a total of 2 [42,46].

In view of the results of this meta-analysis, Pilates seems to have positive efficacy for pain relief and the improvement of functional disorders in patients with chronic low back pain, and the effects of treatment may be maintained for a period. 

  1. Line 281: there is Supplementary Table S2, there should be Supplementary Table 1.

Thank you for your suggestion.

Supplementary Materials:

  1. Table names should be unified with figure names, form example:

Supplementary Figure S1. And Supplementary Table S1.

  1. I suggest changing the table order in Supplemental Materials according to the order in which they appear in the article

Supplementary Table S1. Search strategy for each database.

Supplementary Table S2. Jada scale score of included RTCs.

Supplementary Table S3. The pooled results of sensitivity analyses [MD(95% CI)].

Thank you for your suggestion. We have changed the table names and the table order in the revision of the supplementary material.

  1. Supplementary Table 2: explain below the table what 1, 2, 3 means.

Thank you for your suggestion. We have added table notes below the Supplementary Table S2 in the revision of the supplementary materials.

  1. Supplementary Table 3: The results presented in Table 3 are not clear. I suggest adding an additional column “Search”.

Search engine

Search

Search query

Date of search

Pubmed

#1

Pilates [Title/Abstract]

20-November-22

#3

#1 OR #2

Thank you for your suggestion. We have added column “Search” in Supplementary Table S1 in the revision of the supplementary materials.

Supplementary Table S1. Search strategy for each database.

Search engine

Search

Search query

Date of search

CNKI/VIP/ CBM

#1

主题 (Topic) =普拉提(Pilates) OR普拉提训练(Pilates training)

20-November-22

#2

主题 (Topic) =腰痛(back pain/ low back pain) OR 非特异性腰痛(nonspecific low back pain) OR慢性腰痛(chronic low back pain) OR慢性非特异性腰痛(chronic nonspecific low back pain)

#3

#1 AND #2

PubMed

#1

Pilates [Title/Abstract]

20-November-22

#2

Pilates training [Title/Abstract]

#3

#1 OR #2

#4

low back pain [Title/Abstract]

#5

back pain [Title/Abstract]

#6

low back ache [Title/Abstract]

#7

chronic low back pain [Title/Abstract]

#8

nonspecific low back pain[Title/Abstract]

#9

chronic nonspecific low back pain [Title/Abstract]

#10

chronic nonspecific lumbago [Title/Abstract]

#11

chronic nonspecific lower back pain [Title/Abstract]

#12

chronic nonspecific lumbar pain[Title/Abstract]

#13

non-specific lower back pain [Title/Abstract]

#14

#4 OR #5 OR #6 OR #7 OR #8 OR #9 OR #10 OR #11 OR #12 OR #13

#15

#3 AND #14

Web of Science

#1

TS = (‘Pilates’ OR ‘Pilates training’)

20-November-22

#2

TS = (‘low back pain’ OR ‘back pain’ OR ‘low back ache’ OR ‘chronic low back pain’ OR ‘nonspecific low back pain ’ OR ‘nonspecific low back pain’ OR ‘chronic nonspecific low back pain’ OR ‘chronic nonspecific lumbago’ OR ‘chronic nonspecific lower back pain’ OR ‘chronic nonspecific lumbar pain’ OR ‘non-specific lower back pain’)

#3

#1 AND #2

Databases = SCI-EXPANDED, SSCI, A&HCI, CPCI-S, CPCI-SSH, ESCI

Embase

#1

‘Pilates’ OR ‘Pilates training’

20-November-22

#2

‘low back pain’ OR ‘back pain’ OR ‘low back ache’ OR ‘chronic low back pain’ OR ‘nonspecific low back pain ’ OR ‘nonspecific low back pain’ OR ‘chronic nonspecific low back pain’ OR ‘chronic nonspecific lumbago’ OR ‘chronic nonspecific lower back pain’ OR ‘chronic nonspecific lumbar pain’ OR ‘non-specific lower back pain’

#3

#1 AND #2

EBSCO

S1

Pilates OR Pilates training

20-November-22

S2

low back pain OR back pain OR low back ache OR chronic low back pain OR nonspecific low back pain OR nonspecific low back pain OR chronic nonspecific low back pain OR chronic nonspecific lumbago OR chronic nonspecific lower back pain OR chronic nonspecific lumbar pain OR non-specific lower back pain

S3

S1 AND S2

  1. Citation of publication number 66 is missing.

Thank you for your suggestion. I'm sorry for our negligence. We have corrected the references list in the revision of the manuscript.